# Nutritional Considerations for Bladder Storage Conditions in Adult Females

**DOI:** 10.3390/ijerph20196879

**Published:** 2023-10-03

**Authors:** Barbara Gordon

**Affiliations:** Department of Nutrition and Dietetics, Idaho State University, Meridian, ID 83642, USA; barbaragordon@isu.edu; Tel.: +1-208-373-1904

**Keywords:** interstitial cystitis/bladder pain syndrome, overactive bladder, recurrent urinary tract infection, stress urinary incontinence, nutrition, dietitian, clinical guideline

## Abstract

Background: Clinical guidelines developed by urologic, urogynecologic, and gynecologic associations around the globe include recommendations on nutrition-related lifestyle and behavioral change for bladder storage conditions. This study identified and compared clinical guidelines on three urological conditions (interstitial cystitis/bladder pain syndrome (IC/BPS), overactive bladder, and stress urinary incontinence) affecting adult women. Methods: A three-step process was employed to identify the guidelines. Next, a quality assessment of the guidelines was conducted employing the Appraisal of Guidelines Research and Evaluation (AGREE II) International tool. (3) Results: Twenty-two clinical guidelines, prepared by seventeen groups spanning four continents, met the inclusion criteria. The AGREE II analyses revealed that most of the guideline development processes complied with best practices. The most extensive nutrition recommendations were for women with IC/BPS. Dietary manipulation for the other two storage LUTS primarily focused on the restriction or limitation of specific beverages and/or optimal fluid intake. (4) Conclusion: Clinical guidelines for IC/BPS, overactive bladder, and stress urinary incontinence include nutrition recommendations; however, the extent of dietary manipulation varied by condition. The need to ensure that clinicians are informing patients of the limitations of the evidence supporting those recommendations emerged. Furthermore, given the need to treat nutrition-related comorbid conditions as a strategy to help mitigate these three urological disorders, the value of referral to a dietitian for medical nutrition therapy is apparent.

## 1. Introduction

The role of nutrition in health and disease is well established [1]. Indeed, the need for all clinicians to possess knowledge of nutritional interventions for conditions within their specialty is essential in today’s healthcare delivery system [2]. Many clinical guidelines developed by urologic, urogynecologic, and gynecologic associations around the globe include recommendations on nutrition-related lifestyle and behavioral change for conditions presenting with lower urinary tract symptoms (LUTS) [3,4,5,6,7,8,9,10,11,12,13,14,15,16,17,18,19].

LUTS occur during different phases of bladder functioning, specifically storage, voiding, and post-micturition [20]. During the storage phase, a relaxed bladder slowly fills with urine. Bladder dysfunction during this phase is typically due to spontaneous or provoked involuntary contractions of the bladder wall muscle (detrusor) [20]. Symptoms of storage disorders include urinary tract disorders such as urinary urgency (sudden, urgent need to void, particularly during the daytime), urinary frequency (increased need to void, particularly during the daytime), urinary incontinence (involuntary leakage of urine), nocturia (waking one or more times during the night to void), and pain/discomfort with filling [20]. Disorders associated with those symptoms include interstitial cystitis/bladder pain syndrome (IC/BPS), overactive bladder (OAB), and stress urinary incontinence (SUI) [19,20].

With a focus on women, this review compiled the nutrition recommendations included in clinical guidelines for IC/BPS, OAB, and SUI. An analysis of compliance with the best practices for developing guidelines was conducted, and a comparative analysis of the nutrition recommendations was performed. A review of the strength of the evidence supporting those recommendations is also included. Lastly, insights on the translation of findings into clinical practice are offered.

## 2. Materials and Methods

A three-step process was employed to identify clinical guidelines. Medline/PubMed and Google Scholar were searched from 15 August 2022 to 20 September 2023. Search terms included “guideline” and “name of condition” and “name of the association/society”. Guidelines were also manually collected from the websites of urological, urogynecologic, and gynecologic associations from around the world. Appendix A is a list of association websites accessed for relevant clinical guidelines stratified by continent. Finally, the references of articles were reviewed for candidate guidelines.

Guidelines on the management of the three urological conditions in women published in the past 10 years by urology, urogynecology, and gynecology associations/societies or professional organizations/networks were included. The retrieved guidelines were reviewed, and data extracted on dietary manipulation, including consuming or restricting specific nutritional supplements and the grade assigned for the strength of the evidence supporting those dietary interventions. If a retrieved guideline had been updated but only a summary of the changes published, then the full guideline was retrieved and reviewed; data collection prioritized findings from the most current version. The most current published versions of the guidelines were analyzed and vetted for recommendations. Articles that were not relevant to the clinical care of adult women were excluded. Table 1 details the inclusion and exclusion criteria.

Data were compiled into three tables: (1) a list of clinical guidelines and publication dates stratified by urological condition, (2) an overview of the scoring and grading schemas used to evaluate the level of evidence supporting the recommendations mapped to a standardized schema, and (3) a figure illustrating the strength of evidence (based on the standardized schema) for the nutritional interventions included in each guideline for the three conditions.

A quality assessment of the included guidelines was conducted utilizing the Appraisal of Guidelines Research and Evaluation (AGREE II) International tool [21]. The 23-item AGREE II included six domains: (1) scope and purpose, (2) stakeholder engagement, (3) rigor of development, (4) clarity of presentation, (5) applicability, and (6) editorial independence. In addition, an overall assessment of the quality of the guideline was calculated by dividing the cumulative score by the number of items. Each of the 23 items and the overall guideline assessment rating was scored on a range of 1 (strongly disagree) to 7 (strongly agree). The AGREE II tool also required issuing a recommendation determination: Not recommend, recommend, or recommend with modifications [21]. A copy of the AGREE II tool is included as Appendix A.

## 3. Results

Seventeen organizations spanning four continents (Asia, Australia and Oceania, Europe, and North America) produced clinical guidelines meeting the inclusion criteria:Four of the organizations were urology associations/societies: American Urological Association (AUA), Canadian Urological Association (CUA), European Urological Association (EUA), and the Urological Society of Australia and New Zealand (USANZ) [3,4,5,6,7,8,9,10].Four were urogynecology associations/societies: American Urogynecologic Society (AUGS), British Society of Urogynaecology, International Urogynecologic Association (IUGA), and Urogynaecological Society of Australia (UGSA) [10,11,12,13].Four were gynecology associations/societies: American College of Obstetricians and Gynecologists, Royal College of Obstetricians and Gynaecologists (RCOG), International Society of Psychosomatic Obstetrics and Gynaecology (ISPOG), and the Society of Obstetricians and Gynaecologists of Canada (SOGC) [11,12,14,15].Four guidelines were developed by interprofessional associations: Global IC/PBS Society (GIBS), International Continence Society (ICS), Japanese Continence Society (JCS), and Society of Urodynamics, Female Pelvic Medicine & Urogenital Reconstruction (SUFU) [4,5,17,18].The remaining guideline was developed by the East Asian group of urologists (EAG), an ad hoc group of experts [19].

Seven of the associations published guidelines on more than one of the urologic conditions (AUA, CUA, EUA, ICS, JCS, SUFU) [3,4,5,6,7,8,9,13,17,18]. Of note, two of the guideline development teams produced one document covering multiple urological conditions [17,18]. Seven guidelines were collaboratively developed [4,5,10,11,12,13]:AUA/SUFU on OAB and SUI.AUGS and ACOG on OAB and SUI.BSUG and RCOG on IC/BPS.ICS and IUGA on SUI.SANZ and UGSA partnered on OAB.

A total of 22 guidelines, offered in 16 discrete documents, were included in this review [3,4,5,6,7,8,9,10,11,12,13,14,15,16,17,18,19]. Given the collaboration on the development of some of the guidelines, nine guidelines on IC/BPS, seven on OAB, and six on SUI were analyzed for this review [3,4,5,6,7,8,9,10,11,12,13,14,15,16,17,18,19]. Table 2 provides the details of the publication dates of the guidelines stratified by condition and development group.

The AGREE II analyses revealed that most of the guideline development processes complied with best practices. It was not clear, however, if some of the AGREE best practices were not followed or not reported. For example, the inclusion of all relevant healthcare providers in the development and/or review of the guidelines was often difficult to ascertain; in addition, only seven of the development teams and nine guidelines noted inclusion of the patient representatives [3,5,7,8,9,17].

Most guidelines (18/22) reported on the protocol employed to grade and score the evidence supporting specific recommendations [3,4,5,6,7,8,9,11,12,15,17,18,19]. Seven disparate scoring schemas emerged. Five teams utilized the Grading of Recommendations Assessment, Development and Evaluation schema: AUA/SUFU OAB, EUA

IC/BPS, OAB, SUI, and SOCG SUI guidelines [4,8,9,15]. Four used the (modified) Oxford Centre for Evidence-Based Medicine Levels of Evidence: CUA and ICS, both for IC/BPS and OAB [6,17]. Eight guidelines employed organizational-specific guideline development protocols and evidence-scoring schemas (ACOG/AUGS OAB and SUI, AUA IC/PBS, AUA/SUFU SUI, BSUG/RCOG IC/BPS, and JCS IC/BPS, OAB, and SUI) [3,5,11,12,18]. The EAG schema employed did not “score” the level of evidence supporting recommendations but rather offered four degrees of recommendation, ranging from not recommended to strongly recommended [19]. Four guidelines did not report utilizing this best practice: both GIBS and ISPOG IC/BPS, SANZ/USGA OAB, and ICS/IUGA SUI [10,13,14,16]. Table 3 provides an overview of the scoring and grading schemas stratified by organization and condition. The far-right column maps each grading schema to the standardized version, which includes strong, moderate, weak, clinical principle, expert opinion, recommended but not scored, and no recommendation offered.

### 3.1. Interstitial Cystitis/Bladder Pain Syndrome

Nine IC/BPS guidelines met the inclusion criteria; they were developed by the AUA, BSUG/RCOG, CUA, EAG, EAU, GIBS, ICS, ISPOG, and JCS. [4,5,6,7,8,14,19]. Prevalence rates for IC/BPS varied by country of current residency. On the low end, rates were <1% among women in Taiwan, Korea, Vienna, and Finland (0.04, 0.26, 0.3%, and 0.45, respectively) [3,19]. The EAG reported a 1% prevalence rate among Japanese women. Rates were higher among women in the United States, ranging from 2.7–11% [3,6].

Though an international effort has been made to adopt consistent nomenclature and a uniform definition of IC/BPS, disparities were found for these two items in the nine IC/BPS guidelines. “Interstitial cystitis” was employed by the ICS, ISPOG, and JCS [14,17,18]. The AUA, CUA, and GIBS used the term IC/BPS [3,6,16] and the EAG, hypersensitive bladder syndrome [19]. The BSUG/RCOG guidelines opted for bladder pain syndrome and the EUA primary bladder pain syndrome [12]. Of note, ESSIC (International Society for the Study of Bladder Pain Syndrome) advocates for universal adaptation of the term bladder pain syndrome, or BPS, because interstitial cystitis is not reflective of the underlying pathology of the condition [22]. In addition, BPS is congruent with the International Association for the Study of Pain classification system [15k]. AUA, CUA, and ICS used the ICS definition of IC/BPS, specifically, “an unpleasant sensation (pain, pressure, discomfort) perceived to be related to the urinary bladder, associated with lower urinary tract symptoms for more than six weeks duration, in the absence of infection or other identifiable causes” [3,6,17]. The EAG definition adds that the urinary urge and frequency symptoms were unwavering [19]. The EUA and GIBS guideline definitions also indicate the addition of symptoms such as worsening pain upon filling of the bladder or nocturia [8,16]. The BSUG/RCOG provided several definitions, including the AUA (ICS), EUA, and ESSIC definitions [12]. ISPOG and JCS did not define the term; however, ISPOG classified IC/BPS as a type of chronic pelvic pain and JCS noted its association with LUTS [14,18].

The rounded, overall AGREE II guideline assessment scores for the IC/BPS clinical guidelines ranged from 5 to 7, with 7 being the highest rating. Four of the guidelines did not share whether the development committee included interprofessional team members and/or patient/public representatives [8,16,18,19]. Two of the guidelines did not score the strength of evidence for their nutrition recommendations [8,16]. Also, of note, one of the guidelines was developed by ad hoc group of urological professionals [19]. Appendix A provides a summary of the AGREE II quality analysis for the IC/BPS guidelines.

Eight of the nine guidelines advocated for dietary interventions; ISPOG did not include dietary information in their guidelines [14]. The strength of the evidence supporting nutrition interventions for women with IC/BPS ranged from recommended but not scored to clinical principle to moderate; thus, none of the evidence associating consumption patterns with IC/BPS symptomatology was rated as strong [4,5,6,7,8,19]. Two of the guidelines recognized the value of referral to a registered dietitian for medical nutrition therapy [3,8]. Four guidelines noted the potential value of avoiding individual symptom triggers and recommended an elimination diet to help identify foods and beverages that exacerbate IC/BPS symptoms [3,6,16,17]. Specific details regarding dietary manipulations varied across the eight guidelines. The EUA guidelines only noted the potential value of making dietary recommendations; no specifics regarding those nutritional interventions were provided [8]. Five guidelines advocated for restricting citrus juices and acidic beverages [3,6,12,16,19]. The AUA, BSUG/RCOG, CUA, and GIBS also recommended limiting caffeine [3,6,12,16]. Half of the development teams advocated modulating water intake (increasing/decreasing) based on symptoms [3,6,16,17]. The Canadian and GIBS guidelines also recommended limiting the intake of carbonated drinks/sodas and alcoholic beverages [6,16]. Five organizations (AUA, EUA, ICS, ISPOG, JCS) did not list foods, food substances, or nutrients to restrict [3,8,14,17,18]. The BSUG/RCOG, CUA, EAG, and GIBS, however, provided a list of items to restrict including citrus fruits, spicy foods, tomatoes/tomato products, and other food products; strength of evidence scores ranged from expert opinion to moderate to ungraded, respectively [6,12,16,19]. The AUA offered the clinical principle of the potential benefit of three nutritional supplements—calcium glycerophosphates, nutraceuticals, and phenazopyridine [3]. The CUA scored as moderate strength the evidence supporting the recommendation to restrict vitamin C supplements [6].

An extensive list of comorbid conditions was associated with IC/BPS, including other chronic pain syndromes (fibromyalgia, vulvodynia, Sjogren’s syndrome), gastrointestinal disorders (irritable bowel syndrome, constipation), sexual dysfunction, obstructive sleep apnea, and disrupted sleep [3,6,9,14,16,17,19]. The need to consider these comorbid conditions during both the differential diagnosis process and treatment plan emerged [3,5,8,12,14,18]. For example, increasing fiber and fluid intake to help mitigate non-opioid-induced, chronic constipation [6].

### 3.2. Overactive Bladder

Ten organizations published seven clinical guidelines with nutrition recommendations for managing OAB: ACOG/AUGS, AUA/SUFU, CUA, EUA, ICS, JCS, and SANZ/UGSA [5,6,8,10,11,17,18]. The EUA’s Management of Non-Neurogenic Female Lower Urinary Tract Symptoms, ICS Standards 2023, and JCS’s clinical guidelines for female LUTS included clinical care recommendations for OAB [9,17,18]. Prevalence rates for OAB among adult women range from 9–43%; employing the ICS definition, 12.8% experience the condition [6]. Five of the guidelines used the ICS definition of OAB, specifically urinary urgency, typically presenting with both frequency and nocturia and without urinary urgency or obvious pathology [3,6,8,10]. The ACOG/AUGS Practice Bulletin did not offer a definition [11].

The AGREE II overall, rounded quality scores ranged from 6 to 7. The ACOG/AUGS, AUA/SUFU, CUA, JCS, and SANZ/UGSA guidelines did not note the inclusion of patients in the guideline development process [4,7,10,11,18]. In addition, it was not clear whether the ACOG/AUGS and CUA development teams included interprofessional groups of healthcare providers [7,11]. The SANZ/UGSA guidelines also did not provide author disclosures [10]. Innate methodological variations exist in the ICS 2023 Standards because it is a collection of journal articles [17]. Appendix A provides a summary of the AGREE II quality analysis for the OAB guidelines.

Nutrition recommendations for treating OAB were around fluid intake [5,6,8,10,11,17,18]. Five of the seven guidelines advocated limiting caffeine consumption [4,7,9,10,11,17]. The AUA/SUFU, CUA, and EUA rated the evidence level as moderate, and the SANZ/UGSA did not score the recommendation [4,7,9,10,11,17]. Takahashi stated that JCS guideline developers found no evidence to support this recommendation in the literature [18]. The JCS and SANZ/UGSA guidelines also recommended limiting alcoholic and carbonated beverages; ratings on the strength of the evidence were unscored and weak, respectively [10,18]. Four guidelines advocated modulating water/fluid intake, especially in the evenings [4,7,9,11]. The ACOG/AUGS, AUA/SUFU, and CUA guidelines scored the evidence level as moderate; the EUA scored it as strong [4,7,9,11].

The need to address comorbid conditions that can impact bladder functioning was raised in all seven guidelines [5,6,8,10,11,17,18]. The CUA guidelines included nutrition-related comorbid conditions such as chronic pelvic pain, diabetes, and constipation [7]. JCS included the recommendation to reduce weight among obese women [18]. The SANZ/UGSA and EUA guidelines noted an association between OAB and cardiac conditions and metabolic syndrome [9,10]. The EUA and JCS scored the recommendation to lose weight for women who are overweight/obese to help ameliorate OAB as strong [9,18]. ICS highlighted the increased risk for obstructive sleep apnea syndrome among patients experiencing OAB-related nocturia [17].

Of note, studies positively associated dietary patterns high in sodium and caffeinated tea with nocturia, while those high in fruits and vegetables were negatively associated [23,24]. Furthermore, in a non-randomized clinical trial (*n* = 321 Japanese seniors), Tomohiro et al. found that consumption of ≥7 g of salt daily was associated with nocturia among women [25]. Nocturnal voiding was significantly reduced among those who decreased salt intake (10.7 to 8 g/day). A smaller trial (n = 74) found similar results [26]. Dietary sodium restriction reduced nocturnal urinary frequency (2.5 vs. 1.0 voids, *p* < 0.001) [26]. In a recent review of the literature, however, Alwis et al. concluded that there was insufficient evidence to support dietary recommendations for management of nocturia [23].

### 3.3. Stress Urinary Incontinence

Six guidelines were found for SUI [5,9,11,13,15,18]. Three of the guidelines were collaborations (ACOG/AUGS, AUA/SUFU, and ICS/IGUA), and two were subsets of the LUTS guidelines (EUA and JCS) [5,9,11,13,18]. SOGC published their clinical practice on the evidence supporting conservative care for SUI independently [15]. Prevalence rates for SUI are as high as 49% of women [13]. Though older women appear to be at higher risk, younger women may develop SUI associated with pregnancy and certain athletic activities [13]. The EUA and ICS/IUGA guidelines utilized the ICS definition of SUI—an involuntary loss of urine associated with physical activity [9,13]. SOGC offered the following definition: “the complaint of involuntary leakage on effort or exertion or on sneezing or coughing” [15]. The ACOG/AUGS and JCS guidelines did not define SUI [11,18].

Though the rounded overall AGREE II quality scores were 7, the ACOG/AUGS and AUA/SUFU guidelines did not mention the inclusion of an interprofessional team [5,11]. The JCS guideline did not indicate whether patients were included on the guideline development team [15]. Though an evidence-grading schema was employed for practice recommendations, the SOGC did not score the nutrition recommendations [15]. Appendix A provides a summary of the AGREE II quality analysis for the SUI guidelines.

JCS recommended limiting alcoholic and carbonated beverages but rated the evidence as weak [18]. ACOG/AUGS and ICS/IUGA scored as moderate strength the recommendation to modulate water/fluid intake [11,13]. ACOG/AUGS and SOGC also advised limiting caffeine consumption [11,15]. AUA/SUFU and EUA did not offer any dietary recommendations [5,9].

All six guidelines emphasized the role of overweight/obesity in symptom exacerbation [5,9,11,13,15,18]. ACOG/AUGS, EUA, and JCS scored the strength of evidence supporting the weight loss recommendation as strong, ICS as moderate, AUA/SUFU as expert opinion, and SOGC as ungraded [5,9,11,13,15,18]. The AUA/SUFU guidelines also recommended addressing diabetes, if needed [5]. Figure 1 illustrates the nutrition recommendations stratified by both organization and condition.

## 4. Discussion

A common nutrition recommendation for bladder storage conditions is to modulate the intake of “bladder irritants”. Bladder irritants are typically defined as alcohol, caffeine, citrus juices, soda (regular and diet), and spicy foods [27,28]. However, a lack of rigorous clinical trials demonstrating the influence of diet on these three bladder disorders emerged. Rather, recommendations relied heavily on bench science and observational studies. Indeed, only eight of the nutrition recommendations in the reviewed clinical guidelines were supported by a strong level of evidence [9,11,18]. Six of the eight related to the value of dietary counseling to address obesity to help mitigate OAB and SUI symptomatology; the seventh recommendation was the benefit of restricting evening fluids for women with OAB, and the eight was to provide dietary counseling, as needed, for comorbid conditions for those with IC/BPS [9,11,18]. In addition, disparate expert views of potential nutritional interventions were represented. The need for a global agreement on dietary modifications for women presenting with the urologic conditions reviewed was indicated.

### 4.1. Dietary Bladder Irritants

Evidence from animal and in vitro studies supports the role of bladder irritants on functional changes of the bladder that contribute to the pathophysiology of LUTS. Artificial sweeteners (acesulfame K, aspartame, and sodium saccharin) have been found to have a stimulatory effect on the bladder [29,30]. Elliot et al. found that T1R2/3 sweet taste receptors in bladder urothelium may result in bladder contraction [29]. Dasgupta et al. discovered that via modulation of L-type Ca+2 channels, low concentrations of artificial sweeteners enhance detrusor muscle contractions [30]. Caffeine promotes bladder instability and urinary frequency, bladder detrusor overactivity, and bladder epithelium changes integral in muscle contraction [31]. Even low doses of caffeine affect the normal physiological processes in the bladder, which can promote urinary urgency. Aveno and Cruz found that caffeine affects a capsaicin-sensitive ion channel which regulates pain perception and bladder contractions, suggesting the underlying physiological mechanism by which caffeine consumption triggers IC/BPS symptoms [32]. Citrus products contain varying degrees of ascorbic acid, which has been found to increase both the frequency and intensity of muscle contractions in the bladder due to alteration of neurotransmitters [33]. Carbonated sodas contain ascorbic acid, citric acid, phenylalanine, and colorants. These substances have also been found to enhance bladder muscle contraction [34]. Spicy foods (wasabi, horseradish, mustard, chili peppers) activate sensory nerve endings via TRP channels, producing irritation and inflammation [28,34]; thereby, ingestion of these items may exacerbate bladder pain. Table 4 provides an overview of the bench science supporting the impact of the consumption of certain beverages on the pathogenesis of LUTS and bladder pain.

Additional evidence is based on the findings of observational studies culled from self-report instruments to assess the association between consumption of bladder irritants and LUTS. Though this data collection method introduces the risk of bias (both reliability and validity concerns), these studies offer an understanding of patient experiences. Furthermore, the results of the bench science studies discussed above provide insights into the potential causality of the correlations gleaned from observational studies. A narrative review of the role of diet in managing IC/BPS, which combined the findings of observational studies conducted with thousands of patients, concluded with a proposed algorithm for providing medical nutrition therapy for women with IC/BPS [35]. Of note, the importance of identifying individual food intolerances was emphasized, as well as the finding that most women with IC/BPS are aware of the specific foods and beverages that provoke their symptoms [35]. The Leicestershire MRC Incontinence Study (n = 15,904 community-dwelling adults) also offers epidemiological evidence that nutritional intake appears to be a factor in urinary incontinence [36]. OAB was associated with low intakes of vegetables, protein, vitamin D, and potassium in women, and SUI with a low intake of bread and a high intake of saturated fat, zinc, and vitamin B12 [36]. Bradley et al. conducted a systematic review on the effect of diet (n = 28 articles), fluid intake (n = 21), caffeine (n-21), and alcohol (n = 26) on LUTS. Fluid intake and caffeine consumption were associated with urinary frequency and urgency; however, the level of evidence was low. Inconsistent findings emerged regarding the association between alcohol intake and LUTS [37].

Clinical trials supporting dietary interventions for these three urological conditions are sparse. Three clinical trials were conducted on the effect of dietary intake on IC/BPS. A 3-month, multi-center, randomized, double-blind, placebo-controlled study (n = 231 women) found a 45% favorable response to dietary modifications for managing symptoms [38]. A 1-year, single-center, randomized, controlled trial (n = 30, ≥30 years Japanese females) restricted tomatoes, tomato products, soy, tofu products, spices (pepper, curry, mustard, horseradish), excessive potassium, citrus, caffeine, carbonated products, and citric acid. The intervention significantly alleviated symptoms from baseline to 3 months, and 3 months to 1 year (*p* < 0.05) [39]. A pilot trial (n = 13 women) evaluated an anti-inflammatory diet for interstitial cystitis (AID-IC), which eliminated commonly bothersome foods for those with IC/BPS [33]. The protocol lessened IC/BPS symptoms and improved the quality of life for many of the women in the study [40].

Other trials demonstrate the benefit of reducing sodium intake and controlling fluid consumption as self-care strategies for helping to manage LUTS. Matsuo demonstrated that dietary manipulation can be helpful for managing OAB (n = 98 American adults) [41]. A reduction in dietary salt across 3 months improved voided volumes. Of note, at the end of the study, nearly one quarter (23.9%) of participants no longer presented with OAB [41]. A randomized, prospective, cross-over trial (n = 24 American adults) found that control of fluid intake can also help patients to manage OAB symptoms [42]. A 25% reduction in fluid intake yielded a significant reduction in OAB-related nocturia, as well as urinary urgency and frequency [42]. Clinical trials on diet and SUI focused on the benefits of weight loss for mitigating symptoms [43].

### 4.2. Translation of Findings into Clinical Practice

Nutrition recommendations were found for all three urologic conditions; however, the extent of the dietary modifications varied per condition. The need to ensure that clinicians are informing patients of the limitations of the evidence supporting those recommendations emerged. Only two of the guidelines noted the value of including a registered dietitian as a member of the interprofessional care team—the AUA and EUA IC/BPS guidelines [3,6]. One emerging clinical principle, however, was the need to consider comorbid conditions as a contributing factor. Of note, many of the comorbidities can be treated with medical nutrition therapy (MNT). MNT, which is typically provided by a registered dietitian, is “the…provision of nutrition information/advice (dietary or supplemental) for the assessment/reassessment, diagnosis, intervention or monitoring of a disease or condition, to treat or manage symptoms and/or medical conditions” [1].

MNT for OAB patients with comorbid cardiac conditions, for example, might include dietary modifications increasing daily intake of vegetables, fruits, whole grains, low-fat dairy products, fish, poultry, nuts, and vegetables, while restricting intake of saturated fats, high-sodium foods, added sugars and sugary drinks, and processed and packaged foods [44]. For women with OAB and peripheral edema (depending upon the underlying pathology), fluid and sodium restrictions may be required [44]. A dietitian can also help to address myths and misinformation. Anecdotal evidence, for example, reveals that many women with IC/BPS over-restrict dietary variety [35]. Referral to a registered dietitian can help address these myths [35]. Figure 2 provides an overview of potential MNT for those concomitant conditions stratified by the three urological conditions.

Given that the emerging evidence supporting the nutrition recommendations for these three urologic conditions is generally weak, the need for double-blind clinical trials emerged to evaluate the efficacy and generalizability of dietary manipulation to help manage symptomatology. In addition, a cost and health outcome benefit analysis regarding the engagement of dietitians for the provision of MNT for patients with these conditions requires further study.

A strength of this study was the evaluation of a non-invasive, self-care treatment for bothersome urological conditions which often severely impact a woman’s quality of life. No other studies were found comparing the efficacy of nutrition recommendations for this group of storage-related LUTS. However, the study excluded clinical guidelines developed by professional associations/groups outside of the urological field. Given that these conditions often require the intervention of an interprofessional team of clinicians, the need to further explore and compare clinical guidelines of other professional groups treating these conditions who offer dietary advice to presenting patients, such as urogynecologists, nurses, and physical therapists emerges.

## 5. Conclusions

Twenty-two clinical guidelines, prepared by 17 groups spanning four continents, met the inclusion criteria. The value of employing a standardized schema for scoring and rating future versions of these guidelines emerged. Of note, the most extensive nutrition recommendations were for women with IC/BPS. Dietary manipulation for the other three bladder conditions focused on the restriction or limitation of specific beverages and/or optimal fluid intake. Evidence from clinical trials with small sample sizes also suggested a potential benefit in reducing daily intake of sodium. A lack of strong evidence supporting nutritional recommendations, however, precipitated. The importance of addressing nutritional considerations for comorbidities as a strategy to help manage storage-related LUTS was emphasized and the value of referral to a dietitian for medical nutrition therapy emerged.

## Figures and Tables

**Figure 1 ijerph-20-06879-f001:**
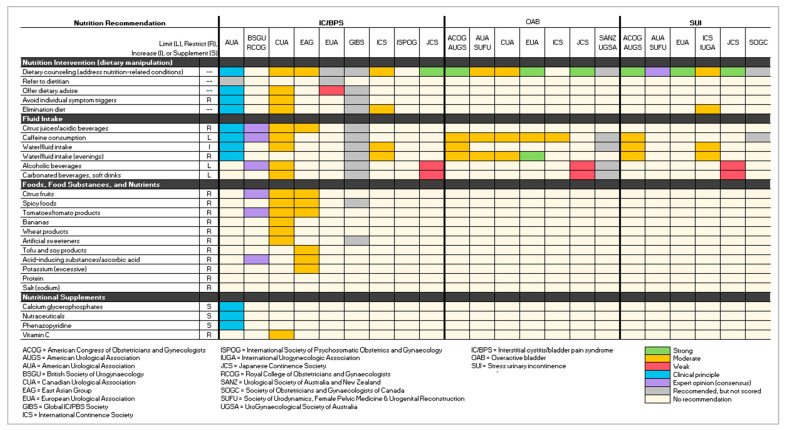
Strength of Evidence of Nutrition Interventions Stratified by Guideline Developer.

**Figure 2 ijerph-20-06879-f002:**
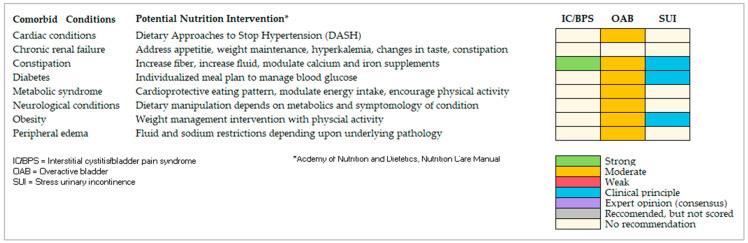
Potential Overlapping Conditions with Evidence-based Medical Nutrition Therapies.

**Table 1 ijerph-20-06879-t001:** Inclusion and Exclusion Criteria Employed to Select Clinical Guidelines.

Inclusion Criteria	Exclusion Criteria
Focused on one or more of the three disorders	Did not provide discrete guidance on specific conditions
Included recommendations relevant to adult females	Recommendations were for children or males
Published in the last 10 years (December 2012–January 2023)	Published prior to December 2012
Developed by urologic, urogynecologic, and gynecologic association, society, or committee	Developed by other associations, societies, or committees
Included relevant treatment interventions	Focused on guidelines for non-treatment issues
Published in the English language	English translation of guideline not found

**Table 2 ijerph-20-06879-t002:** List of Clinical Guidelines and Publication Dates Stratified by Urological Conditions.

Refs.	Acronym	Organizational Developer	IC/BPS	OAB	SUI
[11]	ACOG	American Urogynecologic Society		2014	2014
	AUGS	American College of Obstetricians and Gynecologists			
[3]	AUA	American Urological Association	2022		
[4]	AUA	American Urological Association		2022	
	SUFU	Society of Urodynamics, Female Pelvic Medicine & Urogenital Reconstruction			
[5]	AUA	American Urological Association			2023
	SUFU	Society of Urodynamics, Female Pelvic Medicine & Urogenital Reconstruction			
[12]	BSUG	British Society of Urogynaecology	2016		
	RCOG	Royal College of Obstetricians and Gynaecologists			
[6]	CUA	Canadian Urological Association	2019		
[7]	CUA	Canadian Urological Association		2017	
[19]	EAG	East Asian Group	2015		
[8]	EUA	European Urological Association	2021		
[9]	EUA	European Urological Association		2022	2022
[16]	GIBS	Global IC/PBS Society	2015		
[17]	ICS	International Continence Society	2023	2023	
[13]	ICS	International Continence Society			2017
	IUGA	International Urogynecological Association			
[14]	ISPOG	International Society of Psychosomatic Obstetrics and Gynecology	2015		
[18]	JCS	Japanese Continence Society	2021	2021	2021
[15]	SOGC	Society of Obstetricians and Gynaecologists of Canada			2018
[10]	SANZ	Urological Society of Australia and New Zealand		2018	
	UGSA	Urogynaecological Society of Australia			
Number of Guidelines per Condition			9	7	6
Total Number of Guidelines in Review					22

**Table 3 ijerph-20-06879-t003:** Scoring and Grading Schema for Evidence Supporting Recommendations Mapped to Standardized Schema.

Schema Org: Condition		Grading Schema Employed for Strength of Evidence Rating		Mapped to Standardized Schema	
ACOG/AUGS ACOG/AUGS: OAB, SUI		Level A = Good and consistent scientific evidence Level B = Limited or inconsistent scientific evidence Level C = Clinical practice and expert opinion		Level A = Strong/Moderate Level B = Weak Level C = Clinical principal or Expert Opinion	
AUA AUA: IC/BPS AUA/SUFU: SUI		Grade A/Strong = Rigorous RCT or very strong observational Grade B/Moderate = Weak RCTs or strong observational Grade C/Low = Weak observational Clinical principle (CP) = Widely accepted care practice Expert opinion (EO) = consensus of guidelines panel		Grade A = Strong Grade B = Moderate Grace C = Weak CP = Clinical principle EO = Expert opinion	
BSUG/RCOG BSUG/RCOG: IC/BPS		Grade A = ≥1 Meta-analysis, systematic review or RCT Grade B = High-quality reviews, case-control, cohorts) Grade C = Well-conducted case-control or cohorts Grade D = Case reports, case series, expert opinions √ = Clinical experience		Grade A = Strong Grade B = Moderate Grace C = Weak √ = Clinical principle Grade D = Expert opinion	
EAG EAG: IC/BPS		A = Strongly recommended B = Recommended C = Insufficient evidence for recommendation D = Not recommended		Grade A = Strong Grade B = Moderate Grade C = Weak Grade D = No recommendation	
GRADE AUA/SUFU: OAB EUA: IC/BPS, OAB, SUI SOGC: SUI		High/Strong = Treatment effect confidently same as research findings Moderate = Treatment effect probably close to findings Low/Weak = Treatment effect different from findings Very low = Treatment effect markedly different from findings		High = Strong Moderate = Moderate Low = Weak Very low = Expert opinion	
JCS JCS: IC/PBS, OAB, SUI		A = This action is strongly recommended B = This action is recommended C = There is no clear evidence for recommending this action C1 = Performing the action is not recommended C2 = Not performing this action is recommended D = The action can still be performed Pending = No decision has been made regarding the grade of recommendation		A = Strong B = Moderate C/C1/C2 = Weak D = Expert opinion Pending = No recommendation	
OCEBM CUA: IC/BPS CUA: OAB ICS: IC/BPS, OAB		Grade A (level 1) = Randomized controlled trial Grade B (level 2) = Systematic review of clinical trials, nonrandomized controlled cohort/follow-up study Grade C (level 3) = Case-series, case-control Grade D (level 4) = Expert opinion		Grade A = Strong Grade B = Moderate Grade C = Weak Grade D = Clinical principle/ expert opinion	
Grading Schemas: AUA = American Urological Association EAG = East Asian group of urologists GRADE = Grading of Recommendations, Assessment, Development, and Evaluation JCS = Japanese Continence Society OCEBM = Oxford Centre for Evidence-Based Medicine Levels of Evidence			Organization (Org) Acronyms: ACOG = American College of Obstetricians and Gynecologists AUGS = American Urogynecologic Society AUA = American Urological Association BSUG = British Society of Urogynaecology CUA = Canadian Urological Association EAG = East Asian group of urologists EUA = European Urological Association GIBS = Global IC/PBS Society ICS = International Continence Society ISPOG = International Society of Psychosomatic Obstetrics and Gynecology IUGA = International Urogynecological Association JCS = Japanese Continence Society RCOG = Royal College of Obstetricians and Gynaecologists SANZ = Urological Society of Australia and New Zealand SOGC = Society of Obstetricians and Gynaecologists of Canada SUFU = Society of Urodynamics, Female Pelvic Medicine & Urogenital Reconstruction UGSA = Urogynaecological Society of Australia		

Note: Employment of a formal grading schema was not reported for GIBS (IC/BPS), ISPOG (IC/BPS), SANZ/USGA (OAB), and ICS/IUGA (SUI).

**Table 4 ijerph-20-06879-t004:** Bench Science Findings on the Associations Between Consumption of Caffeine, Citrus Products, and Drinks with Onset of Lower Urinary Tract Symptoms.

Item	LUTS	Mechanism of Action	Source
Artificial sweeteners	Urinary urgency, overactive bladder	Rat and in vitro: Activation of T1R2/3 sweet taste receptors in bladder urothelium may result in bladder contraction	[29]
	Urinary urgency and frequency, nocturia	In vitro: Artificial sweeteners modulate L-type Ca+2 channels provoke detrusor muscle contraction	[30]
Caffeine	Urinary urgency	In vitro: Increased expression of transient receptor potential vanilloid 1 (TRPV1) mRNA in bladder lining mucosa	[31]
	Urinary urgency	Rat model: Affects capsaicin-sensitive ion channel which regulates pain perception and bladder contractions	[32]
	Urinary frequency	Mouse model: Elevated transcription factor c-Fos and nerve growth factor activate neuronal micturition centers	[31]
	Urinary urgency/frequency, incontinence, nocturia	In vitro: Heightened bladder sensory signaling, generating detrusor overactivity	[31]
	Urinary urgency, overactive bladder	In vitro: Affects bladder epithelium, causes changes in the biological pathways integral in muscle contraction	[31]
Citrus foods	Urinary urgency and frequency, incontinence	In vitro: Ascorbic acid increases both the frequency and intensity of muscle contractions in the bladder	[33]
Soda	Urinary urgency and frequency, incontinence	In vitro: Ascorbic acid, citric acid, phenylalanine, and colorants in carbonated sodas disrupt bladder functioning and enhance bladder muscle contraction	[33]
Spicy foods (Wasabi, horseradish, mustard, chili peppers)	Bladder pain	In vitro: Capsaicin and other chemicals found in spicy foods activate sensory nerve endings via TRP channels producing irritation and inflammation	[28,34]

## Data Availability

This is a review of the data available in the peer-reviewed literature. No data were generated during the study. Data generated during the analysis process are included in article figures and tables, and Appendix A.

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
