# Peer review of "Nutritional Considerations for Bladder Storage Conditions in Adult Females"

_ijerph, 2023, doi:10.3390/ijerph20196879_

Round 1

Reviewer 1 Report

The authors should be congratulated for their work. The nutritional assessment and change in nutritional behavior become more and more studied, recently. Moreover, the role of nutrition becomes imperative to manage properly the rehabilitation of cancer patients (PMID: 37642856). Specifically, chronic pathologies could be modified by a correct nutritional assessment and schemes. However, it is not clear the methodology adopted. Specifically, the aim of the study was to compile the nutrition recommendations included in 50 clinical guidelines for IC/BPS, nocturia, OAB, and SUI in women. However, it is not clear (reading lines 44-46), why did you consider nocturia as a condition associated with LUTS. Instead, in my opinion, it will be more appropriate a focus on nocturia which represents the most experienced entity among LUTS and the related ones such as obstructive sleep apnea syndrome (OSAS) (PMID: 37167825). Indeed, high BMI was independently associated with both OSAS and the severity of the disease. Thus a nutritional recommendation may help this subset of patients, improving the overall merit of the manuscript.

Author Response

Thank you for your thoughtful feedback. Major revisions were made to the paper. Based on feedback from both reviewers, the focus is now on IC/BPS, OAB, and SUI. Nocturia is discussed as a symptom of OAB. Please see the attached document for the specifics on the changes made during the revision process.

Reviewer 2 Report

This is a very creative review including recent literature, but major revisions are needed, if this is going to be published. 

- it is beyond of my understanding why you excluded guidelines of non-urological medical scientific organizations and committees. Especially EUGA, IUGA and AUGS recommendations could contribute to your results and conclusions for IC/BPS

- there is no reference on ESSIC guidelines for IC/BPS

- as for ICS or ICI you have to use the most recent edition (2022-2023)

- table 1 does not represent accurately the definition of the investigated situations and should be excluded or replaced

- nocturia must be excluded from the study as it is not a comparable variable. IC/BPS, OAB and SUI are syndromes/ diseases and nocturia just a symptom, actually a possible part of OAB.

Author Response

(The authors gave the same response as above.)

Round 2

Reviewer 1 Report

Now the paper is suitable for publication

Reviewer 2 Report

Accepted in the current revised form.